# Microalgae-Based PUFAs for Food and Feed: Current Applications, Future Possibilities, and Constraints

Anna Santin [1,†], Sergio Balzano [2,3,†], Monia Teresa Russo [1], Fortunato Palma Esposito [2], Maria Immacolata Ferrante [1], Martina Blasio [2], Elena Cavalletti [2] and Angela Sardo [1,4,*]

1   Stazione Zoologica Anton Dohrn, Department of Integrative Marine Ecology, Villa Comunale, 80121 Naples, Italy; anna.santin@szn.it (A.S.); monia.russo@szn.it (M.T.R.); mariella.ferrante@szn.it (M.I.F.)
2   Stazione Zoologica Anton Dohrn, Department of Ecosustainable Marine Biotechnology, Via Acton 55, 80133 Naples, Italy; sergio.balzano@szn.it (S.B.); fortunato.palmaesposito@szn.it (F.P.E.); martina.blasio@szn.it (M.B.); elena.cavalletti@szn.it (E.C.)
3   Department of Marine Microbiology and Biogeochemistry (MMB), Netherland Institute for Sea Research (NIOZ), Landsdiep 4, 1793 AB Texel, The Netherlands
4   Istituto di Scienze Applicate e Sistemi Intelligenti "Eduardo Caianiello" (ISASI), Via Campi Flegrei, 34, 80078 Pozzuoli, Italy
*   Correspondence: angela.sardo@szn.it
†   These authors equally contributed to this work.

**Abstract:** Microalgae are currently considered an attractive source of highly valuable compounds for human and animal consumption, including polyunsaturated fatty acids (PUFAs). Several microalgae-derived compounds, such as ω-3 fatty acids, pigments, and whole dried biomasses are available on the market and are mainly produced by culturing microalgae in open ponds, which can be achieved with low setup and maintenance costs with respect to enclosed systems. However, open tanks are more susceptible to bacterial and other environmental contamination, do not guarantee a high reproducibility of algal biochemical profiles and productivities, and constrain massive cultivation to a limited number of species. Genetic engineering techniques have substantially improved over the last decade, and several model microalgae have been successfully modified to promote the accumulation of specific value-added compounds. However, transgenic strains should be cultured in closed photobioreactors (PBRs) to minimize risks of contamination of aquatic environments with allochthonous species; in addition, faster growth rates and higher yields of compounds of interest can be achieved in PBRs compared to open ponds. In this review, we present information collected about the major microalgae-derived commodities (with a special focus on PUFAs) produced at industrial scale, as well genetically-engineered microalgae to increase PUFA production. We also critically analyzed the main bottlenecks that make large-scale production of algal commodities difficult, as well as possible solutions to overcome the main problems and render the processes economically and environmentally safe.

**Keywords:** microalgae; polyunsaturated fatty acids; marketable products; genetically engineered strains

## 1. Introduction

Agriculture and animal breeding account for about 11% of total greenhouse gas (GHG) emissions, totaling 9.3 GtCO2eq in 2018, 5.3 of which consisted of methane and nitrous oxide derived mostly from livestock (https://www.c2es.org/content/international-emissions; access date: 4 April 2022). The massive GHG emissions from livestock indicate that human consumption of dairy, meat, eggs, and other animal-based products should drastically decline. The environmental impact of animal breeding is also exacerbated by livestock diets, which can occasionally include animal-based products. In particular, fishmeal and fish oils have been supplemented in feed for both terrestrial and aquatic animals for many years. The widespread usage of fish-derived meal in aquaculture and livestock feed is



due to easy availability as fishing bycatches, as well as processing residues of the fishing industry [1]. The nutritional value of animal feed can be improved by adding fish-based products that typically contain high proportions of polyunsaturated fatty acids (PUFAs). PUFA-rich food supplied to terrestrial and aquatic animals contributes to the development of PUFA-rich meat for human consumption. Because of their antioxidant properties, PUFAs exert beneficial effects on health, decreasing the risks of a number of diseases [2–4].

Because of their beneficial effects, PUFAs are also present in a number of nutraceutical products for direct human consumption. Analogously to animal feed, the main source of PUFAs in nutraceuticals are fish oils. Overall, it has been estimated that most PUFA market volume originates from fish oils [5] mostly derived from wild-caught fish, posing serious issues to the sustainability of fishery activities. Fish assimilate PUFAs mostly through their diet by feeding on microalgae or microalgal grazers, such as zooplankton or other PUFA-rich organisms. Alternative PUFA sources include terrestrial plants, microalgae, and some heterotrophic protists. Plants are typically rich in short-chain PUFAs, such as C18:2ω6 (linoleic acid), C18:3ω3 (α-linolenic acid), and C18:3ω6 (γ-linolenic acid), and they are particularly abundant in linseeds and nuts. In contrast, microalgae are particularly rich in two long-chain PUFAs: C20:5ω3 (eicosapentaenoic acid, EPA) and C22:6ω3 (docosahexaenoic acid, DHA). Microalgae and other protists, such as *Thraustochytrids* or heterotrophic dinoflagellates, are the major PUFA producers [6]. Most marine species contain both EPA and DHA in varying proportions, with the former typically more abundant in diatoms, Haptophyta, and green algae and the latter more present in dinoflagellates and *Thraustochytrids*. In addition, Eustigmatophyceae, including *Nannochloropsis* spp., are rich in EPA and do not contain DHA. Within this contest, the food and feed industries are increasingly interested in using and promoting products obtained from plants and microalgae. The latter are receiving considerable attention because of their culturing easiness, their fast growth rates, and the diversity of metabolites the different taxa can produce. Moreover, some microalgae have already received the "generally recognized as safe" (GRAS) status by the Food and Drug Administration (FDA). These include cyanobacteria, such as *Arthrospira* (*Spirulina*) sp., and *Aphanizomenon flos-aquae*; Chlorophyta, such as *Chlamydomonas reinhardtii*, *Auxenochlorella protothecoides*, *Chlorella vulgaris*, *Chlorella protothecoides*, *Dunaliella bardawil*, and *Haematococcus pluvialis*; Rhodophyta, such as *Porphyridium cruentum*; Heterokontophyta, such as *Schizochytrium* sp.; Dinophyta, such as *Crypthecodinium cohnii*; Euglenophyta, such as *Euglena gracilis*; and Bigyra, such as *Ulkenia* sp. [7,8].

Most industrial plants that produce microalgae-derived lipid commodities are based on the employment of open ponds rather than enclosed systems (e.g., photobioreactors). This is mainly due to the lower system and management costs [9–11]. Unfortunately, open systems do not guarantee high levels of purity of algal commodities, as cultures could be affected by bacterial contamination and/or meteoric contributions. Moreover, because of the weekly and seasonal variability in temperatures and light exposure, the maintenance of microalgal cultures in outdoor systems requires phenotypes to achieve fast growth under continuously changing physical conditions. This drastically limits the number of species capable of growing in outdoor ponds. In contrast, the employment of photobioreactors (PBRs)—and thus of enclosed culture chambers—allows for better control of culture conditions, reduces risks of contamination and evaporation of the culture media, and likely allows for massive growth of a wider range of algal species. In indoor systems, the reproducibility of biomass yields and biochemical composition is enhanced by the lack of temperature and irradiance fluctuations. However, the true bottleneck of PBRs are the capital expenditure (CAPEX) and the operating expense (OPEX) associated with their construction and maintenance, respectively [12].

Despite the considerable difficulties related to the economic feasibility of these technologies, several patents exalting the potential of genetically modified microalgae to enhance PUFA yields have been approved in the last decade, paving the way for future applications for the production of microalgae-derived lipids at an industrial scale.

In this review, we provide information about the current market for microalgae-based commodities, genetic transformations aimed at increasing lipid productivities from model microalgal species, the main bottlenecks, and the possible solutions to promote the production and the marketing of microalgal specialties.

## 2. Current Market for Microalgae-Derived Products, with a Special Focus on PUFAs

Microalgae-based products are still niche commodities, the production of which could be encouraged by environmental sustainability policies. Microalgae are suitable alternatives to fossil fuels for the production of biofuels and as agricultural crops for human and animal food applications [13–17]. However, in contrast to research or review articles regarding the microalgal potential for biotechnological purposes, which are easily accessible through scientific search engines, it is difficult to get a precise idea of the true microalgal market and some of its sectors.

In this review, we made a substantial effort to collect information available on microalgal commercialization, with special attention on marketable products derived from microalgal lipids and particularly unsaturated FAs (Table 1 and Figure 1).

Many Countries invest in the microalgal bio-based industry. Such countries are mainly located in North America (United States, Canada, and Mexico), accounting for the majority share of the global ω-3 PUFA market, both in terms of value and volume [18]. However, but investments in the microalgae industry have also been made in Europe (Germany, UK, France, Italy, and Spain), the Asia–Pacific area (China, Japan, India, Australia, and Taiwan), South America, the Middle East, and Africa. The main genera employed for large-scale production are *Arthrospira*, *Dunaliella*, and *Chlorella*, as expected, given the plethora of research articles focused on these species [19–24].

**Table 1.** List of commercially available microalgae-derived commodities. The table includes names and locations of companies, species cultivated and their final products, types of cultivation systems, and links to company websites (n.f.: not found). Websites access date: 15 October 2021.

| Company | Country | Species/Category | Product(s) | Sector | Type of Plant/Growth Condition | Website or Reference(s) |
|---|---|---|---|---|---|---|
| Cyanotech | Hawaii | *Arthrospira* sp. (Spirulina) | Hawaiian BioAstin Hawaiian spirulina | Dietary supplements for human consumption | Open ponds | https://www.cyanotech.com/our-purpose/ |
| Cellana, LLC | Hawaii | Marine microalgae, *Staurosira* sp. | ReNew™ (ω-3 rich oils) | Human and animal food Whole algae enriched with EPA and DHA (as animal feed) | Open ponds | http://cellana.com/ |
| Alltech | United States | *Schizochytrium* sp. | All-G-Rich™ | Dehydrated whole algae for poultry nutrition rich in DHA, biofuels | n.f. | https://www.alltech.com/ |
| TerraVia Holdings, Inc. (formerly Solazyme) | United States | *Chlorella* sp. | Golden Chlorella, AlgaVia (algal powder, food ingredient line recognized as GRAS), AlgaWise (food oils), AlgaPrime DHA (aquaculture) | Dietary supplements (high-value oils and whole-algae ingredients) for human and animal (aquaculture) consumption | Heterotrophic growth in stainless-steel containers | https://www.solazyme.com/ |
| Algenol | United States | *Arthrospira* sp. (spirulina) (other strains available) | Whole algae or protein isolates, phycocyanin | Food and food colorants | Open ponds | https://www.algenol.com/ |
| Omega Tech | United States | *Schizochytrium* sp. | DHA Gold (oil) | Food supplements | n.f. | [25] |

**Table 1.** *Cont.*

| Company | Country | Species/Category | Product(s) | Sector | Type of Plant/Growth Condition | Website or Reference(s) |
|---|---|---|---|---|---|---|
| Martek Biosciences Corporation | United States | *Crypthecodinium cohnii* | ω-3, ω-6, ARA | Food, beverages, dietary supplements, and early-life nutrition | n.f. | https://www.dsm.com/corporate/home.html https://www.linkedin.com/company/martek-biosciences/ |
| GCI Nutrients | United States | *Chlorella, Arthrospira* sp. (Spirulina) | DHA3Sure™ DHA Algae 35% oil complex, organic Chlorella, organic broken-cell Chlorella, | Nutrients | n.f. | https://gcinutrients.com/ |
| AZBIO | United States | Marine microalgae | AlgaeBio Omega-3 Origins™ | EPA and DHA for food and feed | Autotrophic growth | https://www.azbio.org/tag/algae-biosciences |
| Arizona Algae Products, LLC | United States | Marine microalgae | Protein+omega3 powder, EPA extract | Food, dietary supplements, and wellness products | Photobioreactors | https://www.azalgae.com/ |
| Taau Australia Pty Ltd. | Australia | *Arthrospira* sp. (spirulina) | Tabs and powder | Human consumption | Open ponds | https://www.taau.com.au/company.html |
| Photonz Corporation | New Zealand | Marine microalgae | EPA | Aquaculture (fish oil replacement purposes), pharmaceuticals | Fermentation processes | https://pureadvantage.org/photonz-corporation/ |
| Blue Biotech International GmBH | Germany | *Nannochloropsis, Haematococcus* | Algal concentrate, frozen paste, freeze-dried cells, phycocyanin, astaxanthin, microalgae powder | Hatchery, feed | Photobioreactors | https://www.bluebiotech.de/com/index.html |
| Nutrinova | Germany | *Ulkenia* sp. | DHActive™ | Aquaculture | Fermentation processes (80 m$^3$) | [25] |
| Corbion N.V. | Netherlands | *Schizochytrium* sp. | AlgaPrime™ | Aquaculture, pet, and livestock industries | Fermentation processes | http://www.corbion.com |
| Veramaris | Netherlands | Marine microalgae | Veramaris® Pets (algal oil rich in EPA and DHA; GRAS product) | Dog food | n.f. | https://pets.veramaris.com/ |
| Duplaco | Netherlands | *Chlorella* | Powder, food ingredients, and supplements | Food | Photobioreactors and fermenters | https://duplaco.com/ |
| Neoalgae | Spain | *Spirulina, Chlorella* | Powders and capsules | Food | Photobioreactors | https://neoalgae.es/?lang=en |
| AlgaEnergy | Spain | *Arthrospira* sp. (spirulina) | Dietary supplements, feed for aquaculture, biostimulants for agriculture | Food and feed | Outdoor open ponds and outdoor photobioreactors | https://www.algaenergy.it/ |
| AlgAlimento | Spain | *Tetraselmis* sp., *Dunaliella salina, Arthrospira* sp. | Powder for antioxidant food supplements and cosmetics | Food | Outdoor open ponds | http://www.algalimento.com/ |
| Fitoplancton marino | Spain | *Tetraselmis chuii* | Cosmetics, food supplements, feed as Easyreefs® and Easyalgae® | Food, feed, and cosmetics | Open ponds and photobioreactors | http://www.fitoplanctonmarino.com/index.html |
| Microphyt | France | *Arthrospira* sp. (spirulina), *Haematococcus pluvialis* | Powders and pastes, dietary supplements | Food, feed, and cosmetics | Photobioreactors (indoor and outdoor systems) | http://www.microphyt.eu/ |

**Table 1.** *Cont.*

| Company | Country | Species/Category | Product(s) | Sector | Type of Plant/Growth Condition | Website or Reference(s) |
|---|---|---|---|---|---|---|
| Inalve | France | Biofilm-forming microalgae | Powders and pastes for food and feed supplements (e.g., FEAL™) | Food and feed | Photobioreactors | https://www.inalve.com/ |
| FermentAlg | France | *Schizochytrium* sp. | Powder and pastes for food supplements (e.g., DHA-ORIGIN ™) | Food | Fermentation process | https://www.fermentalg.com/ |
| Algenuity | United Kingdom | *Chlorella vulgaris* | Food ingredients | Food | Photobioreactors | https://www.algenuity.com/ |
| AllMicroalgae | Portugal | *Chlorella vulgaris, Tetraselmis chui, Nannochloropsis oceanica, Scenedesmus obliquus* | Powders and pastes | Food and feed | Photobioreactors (outdoor systems) | https://www.allmicroalgae.com/en/ |
| A4F-Algafuel | Portugal | *Dunaliella salina* (pilot), *Lobosphaera incisa, Prorocentrum casubicum, Raphidonema* sp., | Food colorants, ω-6 food additives for children, and dietary supplements | Food and feed | Open ponds and photobioreactors | https://a4f.pt/pt |
| Norsan | Norway, Germany | *Schizochytrium* sp. | ω-3 oil | Food supplements | n.f. | https://www.norsan-omega.com/ |
| Algaria Spireat | Italy | *Arthrospira* sp. (spirulina) | Snack and dietary supplements | Food supplements | n.f. | https://spireat.it/ |
| Biospira Srl | Italy | *Arthrospira* sp. (spirulina) | Capsules, powders, flakes | Food supplements | Monitored tanks that isolate the algae from the external environment | https://www.biospira.it/ |
| Biosyntex | Italy | *Arthrospira* sp. (spirulina), *Haematococcus pluvialis* | Food ingredients, beverages, nutraceuticals, cosmetics, feed for aquaculture, and biostimulants for agriculture | Mainly food, feed, and biostimulants | Indoor and outdoor photobioreactors | www.biosyntex.com/ |
| Micoperi Blue Growth | Italy | *Phaeodactylum tricornutum, Arthrospira platensis, Euglena gracilis* | Capsules, powders | Food supplements | Photobioreactors and indoor ponds | http://www.micoperibg.eu/?page_id=101 |
| TOLO Green Srl | Italy | *Arthrospira* sp. (spirulina) | Food and feed supplements, biostimulants | Food, feed, and agriculture | Open ponds and monitored tanks isolated from the external environment | https://www.tologreen.it/ |
| Alghitaly | Italy | *Arthrospira* sp. (spirulina) | Powders and pastes | Food and feed supplements | Outdoor photobioreactors | https://www.alghitaly.it/ |
| Brevel | Israel | n.f. | Natural salmon feed | Feed | Indoor photobioreactors | https://brevel.co.il/ |
| Algatech | Israel | *Nannochloropsis* sp., *Porphyridium cruentum* | Food supplements | Food | Outdoor photobioreactors | https://www.algatech.com/ |
| Chlorella Industry Co., Ltd. | Japan | *Chlorella* | Tablets, extracts for food ingredients | Food and feed | Open ponds | https://www.chlorella.co.jp/ |
| Yaeyama Syokusan | Japan | *Chlorella* | Powder, tablets, food, and feed supplements | Food and feed | Open ponds | https://www.yaeyamachlorella.com/ |
| Japan Algae | Japan | *Arthrospira* sp. (spirulina) | Powder and tablets for food ingredients and supplements | Food | Open ponds | http://www.sp100.com/ |

**Table 1.** *Cont.*

| Company | Country | Species/Category | Product(s) | Sector | Type of Plant/Growth Condition | Website or Reference(s) |
|---|---|---|---|---|---|---|
| Taiwan Chlorella Manifacturing Co., Ltd. | Taiwan | *Chlorella sorokiniana* | Tablets and food supplements | Food | Open ponds | http://www.taiwanchlorella.com/ |
| Far East Microalgae Industries Co., Ltd. | Taiwan | *Chlorella, Arthrospira* sp. (spirulina) | Powder, dietary supplements, skin care systems, aquaculture feeds | Food, feed, and cosmetics | Open ponds | http://www.femico.com.tw/ |
| Bluetec Naturals Co., Ltd. | China | *Chlorella, Arthrospira* sp. (spirulina) | DHA-rich powder and oil | Food supplement | Open ponds monitored and isolated from the external environment | https://www.bestphycocyanin.com/ |
| Tianjin Norland Biotech Co., Ltd. | China | *Chlorella, Spirulina, Haematococcus pluvialis* | Powder, oils, and tablets | Food | Open ponds | http://www.norlandbiotech.com/ |
| Hangzhou OuQi Food co., Ltd. | China | *Spirulina, Chlorella, Dunaliella salina* | Organic Spirulina, organic Chlorella, broken Chlorella, organic Dunaliella | Food ingredients and supplements | Indoor and outdoor systems | http://www.onlygreen.cn/webEn/LM_about%20us |
| Shaanxi Rebecca Bio-Tech Co., LTD | China | *Dunaliella salina, Haematococcus pluvialis* | Powder, extracts | Natural food | Fermentation process | http://it.rebeccabio.com/ |
| Seagrass Tech Private Limited | India | *Dunaliella salina, Chlorella salina, Arthrospira subsalsa* | SeaCarotene®, SeaLipro®, SeaProtein® | Food | n.f. | https://seagrasstech.com/company-overview/ |
| E.I.D.-Parry Limited | India | *Arthrospira* sp. (spirulina) | Organic Chlorella, spirulina, Phycocyanin, natural astaxanthin | Nutraceuticals | n.f. | https://www.eidparry.com/ |

An accurate analysis available online established that global microalgae market is expected to grow, reaching USD 4049.6 million by 2026, from USD 3391.5 million in 2020, at a compound annual growth rate (CAGR) of 3.0% between 2020 to 2026 (https://southeast.newschannelnebraska.com/story/46278524/microalgae-market-size-in-2022-30-cagr-with-top-countries-data-which-product-segment-is-expected-to-garner-highest-traction-within-the-microalgae; access date: 15 June 2022). The microalgal market segments are food and beverage, animal feed, pharmaceuticals and nutraceuticals, and personal care.

The global ω-3 market was estimated to be worth USD 2.49 billion in 2019 and is expected to expand at a compound annual growth rate (CAGR) of 7% between 2020 and 2027 [26]. FAO statistics and, in particular, FAO report n° 978/2010 [27] reported a market size of EPA and DHA of USD 300 million and 1.5 billion and prices of USD 0.2–0.5/gm and USD 18–22 /gm, respectively [18]. The driving forces for the expected growth of the FA market and, in particular, ω-3, are mainly related to the demographic increase, the effectiveness of PUFAs against cardiovascular diseases (EFSA 2010), and the growing market for functional ingredients [26].

However, information on the main microalgae-derived products with commercial value is still scarce, although our research suggests that they are mainly employed as food supplements and animal feed. In our opinion, this could be due both to a real necessity of healthy and viable alternatives to animal-derived nutritional products and to the long times required for preclinical and clinical trials, which do not encourage the employment of these commodities as pharmaceuticals. Preclinical and clinical trials to demonstrate the safety and the beneficial effects of microalgal products can last approximately ten years [28].

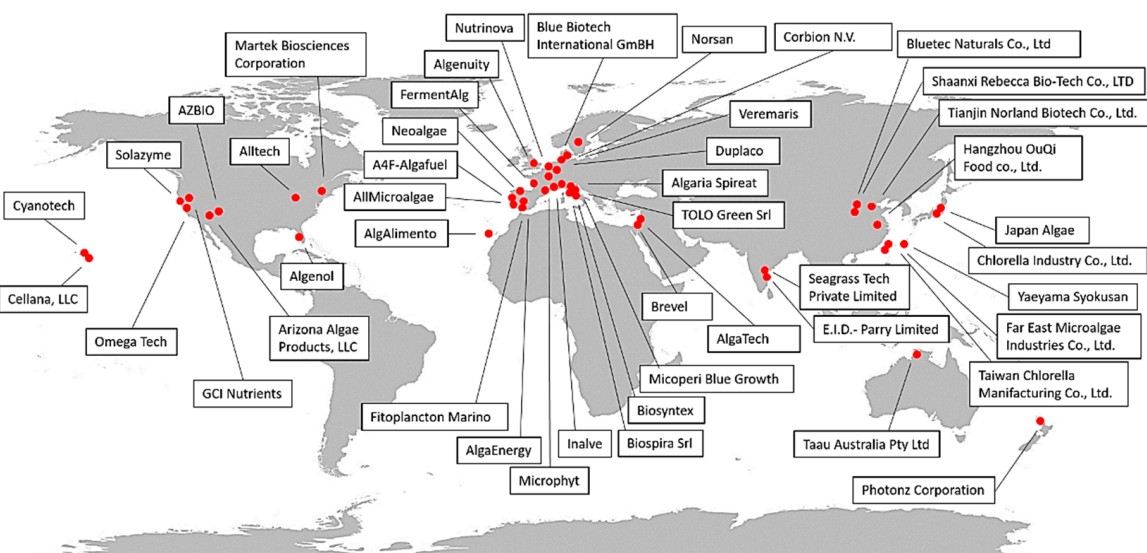

**Figure 1.** Map representing the main companies that produce PUFA-rich microalgae and other microalgae-derived commodities.

In Table 1, we summarized the main information regarding industrial societies with a website or cited in literature papers that produce microalgae-based products for food and/or feed. Companies that commercialize microalgal products are mostly based in the US, EU, China, and Japan (Figure 1). Although this survey does not cover the whole market of microalgal nutraceuticals due to the difficulty of finding exhaustive information, it seems clear that *Arthrospira* is, by far, the most employed microalgal genus for the industrial sector of food supplements for both human and animal nutrition. Given the large number of culturable microalgal species worldwide, few species are commercially produced. This is probably due to the strict food safety regulations and difficulty of adapting algae for large-scale production. Several companies also produce other microalgal commodities for different market sectors. For example, Cellana LLC produces biofuels from microalgae, whereas Photonz Corporation exploits microalgae for pharmaceutical purposes, producing high-purity EPAs to treat cardiovascular diseases.

Microalgal products are mainly commercialized as dried algae, mostly obtained from *Chlorella* and *Arthrospira* spp. Dried algae are rich in vitamins and are usually sold as dietary supplements and specialty products, such as ω-3 FAs, astaxanthin, β-carotene, and phycocyanin, that can be added to food and feed to improve their nutritional value [29].

Regarding industrial plants for microalgal culture exploitation, the information gathered in Table 1 clearly shows that open ponds are the most widespread cultivation systems, being less expensive than enclosed systems. Photobioreactors require a high power supply to maintain constant growth conditions but guarantee low risks of contamination.

## 3. Microalgae-Based Genetic Engineering Technology for PUFA Production: Most Relevant Patents

Different strategies have been studied in order to improve PUFA production from microalgae in recent decades. These include the screening and selection of more productive species and strains, as well as the modulation of growth conditions, such as nutrients, light, or $CO_2$ [30]; and the most recent and most promising strategy to increase PUFA production is represented by genetic engineering. Thanks to the rapid development of high-throughput sequencing technologies and genetic engineering techniques, several microbes, including microalgae, have been successfully transformed. Manipulation of metabolic microbial pathways can often result in transformed genotypes with valuable biotechnological properties, such as increased PUFA content, resulting in a dry biomass more suitable for commercialization compared to the same dry biomass harvested from the wild-type geno-

type. The number of patents released annually has increased sharply in the last 10 years, and this demonstrates the need to understand the molecular mechanisms that regulate PUFA production in microalgae and to exploit this knowledge to manipulate microorganisms in order to find more sustainable solutions.

Despite the lack of solid regulation in many countries and the large gap between production costs and ultimate benefits, genetic engineering research groups have identified interesting strains and genes that can be modified to produce more PUFAs.

As regards genetic engineering, the model microalgae are the freshwater green algae *Chlorella* and *Chlamydomonas*, the eustigmatophycean *Nannochloropsis*, and the diatom *Phaeodactylum*. These species have been studied for many years, their genomes are fully available, and transformation techniques have been developed in order to easily obtain transgenic strains. However, in recent years, the genomes of many other microalgal species have been sequenced and studied in order to broaden knowledge and identify new opportunities for biotechnological applications. An example is the diatom *Cyclotella*, which has attracted increasing interest for biofuel development because of its high TAG content [31]. Moreover, new projects are ongoing to sequence a large number of microalgal genomes (https://jgi.doe.gov/csp-2021-100-diatom-genomes/; access date: 8 February 2022). In addition, information on the most frequently expressed genes and the resulting predicted proteins is available in transcriptome datasets, such as EukProt [32] and MMETSP [33,34].

Various transformation techniques are currently available and have been successfully applied to different microalgal species, from electroporation to microparticle bombardment [30,35]. Furthermore, various editing approaches are possible, for example, overexpressing genes encoding for enzymes that represent critical steps of PUFA production pathways or expressing a gene in another organism (called heterologous expression).

The overexpression of endogenous genes is particularly useful when encoded enzymes are directly involved in fatty acid biosynthesis, with the aim of increasing the final metabolite production (Table 2). For example, beta-ketoacyl-acyl carrier-protein synthase (KAS) and thioesterases (FAT or TE) were overexpressed in the microalgae *Chlorella* or *Prototheca* spp. to increase the production of C8 and C16 fatty acids [36]. Alternatively, genes coding for glycerol-3-phosphate acyltransferase (GPAT) or diacylglycerol acyl-transferase (DGAT), both involved in triacylglycerols (TAGs), can be overexpressed to alter lipid equilibrium within cells, increasing the production of TAGs and/or fatty acids in the diatom *Phaeodactylum tricornutum* [37,38], the green microalga *Chlamydomonas reinhardtii* [39] or the oleaginous alga *Nannochloropsis oceanica* [40]. Other interesting examples of gene overexpression in microalgae are represented by transcription factors, such as bHLH2 in *Nannochloropsis salina* [41] and DOF in *C. reinhardtii* [42]; these two studies demonstrated how acting on a single transcription factor can lead simultaneous modification of multiple steps of biosynthetic pathways, altering global cellular metabolism and increasing lipid production.

On the other hand, heterologous expression is particularly productive when an organism can be easily cultured in the laboratory, achieving fast growth under standard culturing conditions, although resulting in low content of specific metabolites, such as PUFAs. PUFA content is selected and transformed with a transgene, which increases its PUFA production. Often, these model organisms are bacteria [43] or yeasts [44], for which extraction and purification techniques are better defined and less costly than for microalgae, as well as plants, which can be used as PUFA-enriched, plant-based biomass without requiring many purification steps [45]. Genes encoding FAT, GPAT, or DGAT can also be expressed through heterologous expression [43,46] (Table 2). Other interesting studies have involved genes encoding elongases and desaturases of the PUFA biosynthetic pathway; manipulation of these genes can lead to the alteration of PUFA proportions, increasing the levels of PUFAs with more interesting nutraceutical applications, such as EPA and DHA [44]. Moreover, Hu et al. [47–49] expressed *C. ellipsoidea* nuclear factor Y (NF-Y) into plant *A. thaliana*; these factors, which were firs described here for *C. ellipsoidea*, lead to increased biomass and fatty acid production in plants (Table 2).

**Table 2.** List of patents of microalgae that have been genetically modified to enhance their PUFA production. ACS: ADP-forming acetyl-CoA synthetase; bHLH2: basic helix–loop–helix transcription factor 2; BTA: betaine lipid synthase; CAO: Chlorophyllide a oxygenase; DES: desaturase (the number near the Greek letter Δ indicates that the double bond is created at a fixed position from the carboxyl end of a fatty acid chain); DGAT: diacylglycerol acyltransferase; DOF: DNA binding with one finger-type transcription factor; FAB: stearoyl-acyl carrier protein desaturase; GAPDH/GPDH: glyceraldehyde-3-phosphate dehydrogenase; GPAT: glycerol-3-phosphate acyltransferase; KAS: beta-ketoacyl-acyl carrier-protein synthase; NADK: NADH kinase; NF: nuclear factor; TE/FAT: thioesterase/acyl-ACP thioesterase.

| Transformation | Gene | Species | Results | Patent | Submission Date | Reference |
|---|---|---|---|---|---|---|
| Overexpression | KASI-KASIV-FATA-FATB | *Chlorella* or *Prototheca* spp. | +C8-C16 FAs | US 20180230442 A1 | July 2014 | [36] |
| Overexpression | FAB2 | *C. reinhardtii* | +FAs | KR20140005001 A | July 2012 | [50] |
| Overexpression | GPAT | *P. tricornutum* | +TAGs | CN105219649 A | October 2015 | [37] |
| Overexpression | DGAT1 | *P. tricornutum* | +TAGs | US2014196177 A1 | November 2010 | [38] |
| Overexpression | DGAT | *N. oceanica* | +TAGs, +PUFAs | CN110305883 A | March 2018 | [40] |
| Overexpression | DGAT2-5 | *C. reinhardtii* | +biomass, +neutral lipids | CN102321642 A | September 2011 | [39] |
| Overexpression | GAPDH | microalgae | +biomass, +lipids | WO2015105233 A1 | January 2014 | [51] |
| Overexpression | CAO | *C. reinhardtii* | +biomass, +lipids | KR101855739 B1 | June 2017 | [52] |
| Overexpression | bHLH2 | *N. salina* | +lipids | KR20160142024 A | June 2015 | [41] |
| Overexpression | DOF | *C. reinhardtii* | +lipids | CN105755034 A | March 2016 | [53] |
| Heterologous expression | 8 nucleotide sequences | from *C. bastropiensis* to bacteria and mi-croalgae | +FAs | JP2017127278 A | January 2016 | [54] |
| Heterologous expression | Acyl-ACP TE | from *E. siliculosus* to bacteria and microalgae | +lipids | JP2016007154 A | June 2014 | [43] |
| Heterologous expression | DGAT1 | from *C. ellipsoidea* to yeasts, plants or microalgae | +FAs | CN103397007 A | July 2013 | [46] |
| Heterologous expression | GPDH | from *C. ellipsoidea* to yeasts, plants or microalgae | +FAs | CN104357415 B | October 2014 | [55] |
| Heterologous expression | Sucrose invertase | bacteria, yeasts and microalgae | +lipids | EP3546588 A3 | June 2007 | [56] |
| Heterologous expression | Δ12-DES | from *P. viridis* | +LA, +EPA | CN104388442 A | November 2014 | [57] |
| Heterologous expression | Δ8-DES | from *P. viridis* | +DHA, +EPA | CN104293769 A | October 2014 | [58] |
| Heterologous expression | Δ4-DES | from *E. sphaerica* to yeast *P. pastoris* | function confirmed | CN102559710 B | June 2011 | [44] |
| Heterologous expression | NADK3 | from plant *A. thaliana* to *C. pyrenoidesa* | +FAs | CN105316358 A | October 2015 | [59] |
| Heterologous expression | NF-YA | from *C. ellipsoidea* to plant *A. thaliana* | +44.9% weight, +22.4% FAs | CN107936098 A | December 2017 | [47] |
| Heterologous expression | NF-YB | from *C. ellipsoidea* to plant *A. thaliana* | +51% weight, +11.2% FAs | CN108003226 A | December 2017 | [48] |
| Heterologous expression | NF-YC | from *C. ellipsoidea* to plant *A. thaliana* | +44.9% weight, +15.4% FAs | CN108101973 A | December 2017 | [49] |
| Silencing | ACS1-ACS2 | *C. reinhardtii* | +fatty acid excretion | CN105647957 A | January 2016 | [42] |
| Silencing | BTA1 | *Chlamydomonas* sp. | −glycolipids, 2.5-3X TAGs | KR101893522 B1 | April 2017 | [60] |
| Polygene co-silencing | Carbon metabolism genes | *C. reinhardtii* | +biomass, +lipids | CN110564623 A | September 2019 | [61] |

In other cases, silencing is a useful strategy when targets are genes involved in PUFA catabolism or alternative pathways [30]. For example, Huang et al. [42] silenced the acetyl-CoA synthetase (ACS) of *C. reinhardtii*, increasing fatty acid excretion (Table 2). Although some interesting applications of silencing techniques have been reported, this approach is often preferred to characterize gene function in basic research and less used for biotechnological applications. With the expansion of this field in recent years, the next step is the application and regulation of such products in the market.

Although the most exploited genera for large-scale production ensure medium to high biomass and lipid productivities, their genetically engineered clones could significantly increase the yields of microalgae-derived commodities with respect to their wild types, as shown in Table 3.

**Table 3.** Comparison between biomass and lipid productivities of the main species with a commercial value and their genetically modified clones. WT: wild type; GMO: genetically modified organisms.

| Species | WT Biomass | WT Lipid Productivity | Reference | Number of GMO Strains | GMO Lipid Yields | Reference |
|---|---|---|---|---|---|---|
| *Arthrospira platensis* | 0.14 g/L/day | 14.37 mg/L/day | [62] | - | - | |
| *Chlamydomonas malina* and *Chlamydomonas reinhardtii* | 0.53 g/L/day (*C. malina*); 0.014 g/L/day (*C. reinhardtii*) | Total lipid 161.3 (*C. malina*); 10.9 mg/L/day (*C. reinhardtii*) PUFAs 85.4 mg/L/day (*C. malina*) | [63,64] | 26 (of which 4 are patented) | Total lipids ~3.2X more than *C. reinhardtii* WT (~118.3 µg/mL culture) | [65] |
| *Scenedesmus obliquus* | 0.16 g/L/day | 26.77 mg/L/day | [66] | 4 | Biomass ~17% higher than *S. obliquus* WT (~234.3 mg/L/day) and lipid productivity ~2.2X more than WT (~42.4 mg/L/day) | [67] |
| *Chlorella vulgaris Chlorella pyrenoidosa* | 0.73 mg/L/day (*C. vulgaris*) and 0.34 g/L/day (*C. pyrenoidosa*) | 204.91 mg/L/day (*C. vulgaris*) and 66 mg/L/day (*C. pyrenoidosa*) | [68,69] | 6 (of which 2 are patented) | Neutral lipid content ~3.2X and total PUFAs > 34% higher than *C. pyrenoidosa* WT | [70] |
| *Tetraselmis* sp. | 0.30 g/L/day | 43.4 mg/L/day | [71] | - | - | |
| *Dunaliella salina* | 1.03 g/L/day | 0.24 mg/L/day | [72] | 3 | Oil content ~13% higher than *D. salina* WT (~25%) | [73] |
| *Nannochloropsis oceanica* | 0.427 g/L/day | 39.6 mg/L/day | [74] | 11 (of which 2 are patented) | Lipid production 110.6% higher than *N. oceanica* WT (~1.15 g/L) and TAG 148.6% higher than WT (~0.80 g/L) | [75] |
| *Phaeodactylum tricornutum* | 0.254 g/L/day | 99.23 mg/L/day | [76] | 35 (of which 2 are patented) | TAG ~45X more than *P. tricornutum* WT | [77] |
| *Schizochytrium* sp. S31 | 0.81 g/L/day | 100.74 mg/L/day | [78] | 3 | Total lipid yield ~39.6% higher than *Schizochytrium* sp. WT (~110.5 g/L) | [79] |

## 4. Main Bottlenecks for PUFA-Derived Microalgae for Large-Scale Production and Possible Solutions

Although commercial-scale cultivation of microalgae is quickly improving, production costs of microalgal biomass are still far from competitive. The main reasons behind the high production costs are related to the low yields of PUFAs within microalgal biomass [80], which may depend on the cell growth and on the extraction methods. Novel growth strategies aimed at achieving economic viability of mass culturing foresee the employment of waste carbon sources in heterotrophically growing cultures [81] in order to recycle and minimize reject goods, as well as the improvement of harvesting and dewatering technologies, which account for up to 20–30% of the whole biomass cost [71,82–84]. However, the high

costs related to artificial illumination and energy requirements of air and/or gas sparging inside photobioreactors actually make their use economically unsustainable; thus, open ponds rather than enclosed chambers are more feasible as cultivation systems. On the other hand, open ponds are more prone to contamination compared to PBRs, potentially compromising growth and final product purity. The latter aspect should not be neglected in the food and feed sector, as it represents one of the main issues with respect to microalgal products commercialized for human and, to a lesser extent, animal consumption.

The patents described in Table 2 with the aim of optimizing the yields of highly valuable lipids underline the urgent need to grow microalgae in enclosed systems to increase their productivity; in addition, for the cultivation of allochthonous species, enclosed systems also minimize risks of biological contamination of aquatic environments compared to open ponds. Striking a balance between cost-effectiveness and cutting-edge technologies is crucial from the perspective of large-scale production of microalgae-derived commodities.

Another strategy to reduce costs of microalgal mass culturing is based on minimizing the amount of waste, thus exploiting a larger fraction of the microalgal biomass generated by the whole production process. In addition to PUFAs, other fractions, such as TAGs, sterols, carotenoids, and alcohols, can be exploited from the lipid pool. Moreover, both the protein and carbohydrate fractions of microalgal biomass can be exploited to obtain other microalgae-derived commodities. Microalgae are a potential source of several protein ingredients, such as whole-cell proteins, protein concentrates, isolates, hydrolysates, and bioactive peptides [85], as well as a source of recombinant proteins mainly derived from chloroplasts [86]. Carbohydrates, on the other hand, are mainly used as source of bioenergy [87] and, recently, as an eco-sustainable source of prebiotics [88]. In diatoms, frustules could be recovered and used for biomedical applications—e.g., as drug carriers and matrices for tissue regeneration [89–91], as well as for pollutant removal [92]. Finally, alternative methods for PUFAs extraction foreseeing the use of a low or null solvent amount could be appreciable from the perspective of an environmentally and economically sustainable route. Conventional extraction methods with organic solvents are still the main techniques used to extract lipids for microalgae [93,94] due to the high percentages of final products obtained, but they require more subsequent purification steps to remove as much solvent as possible from the final product. Therefore, other methods, such as enzymatic disruption, ultrasonication, microwave methods [95,96], or supercritical fluid extraction [97–99], should be taken into account and optimized to reach a good compromise between lipid yield costs and environmentally safe procedures.

One of the main challenges is represented by the adaptation of legislations concerning the production and the market introduction of microalgal products, in particular with regard to genetically engineered strains for food and feed applications. European countries seem to be generally more reticent than other countries to consider genetically modified organisms (GMO) as nutraceuticals. The US Department of Agriculture (USDA) has decided that all the regulations should relate to the final product or the organism itself rather than the process. This means that the USA tends to rule in favour of gene-edited organisms with regard to approval for growth or entry into the food chain [100]. This is in contrast to the EU and the UK, where current policy and GMO guidelines continue tightly control plants, animals, fungi, and microorganisms modified by gene editing, with severe limitations on their applications. For example, the new European regulatory framework based on the precautionary principle is composed of Directive 2001/18/EC on the deliberate release into the environment of GMOs, taking into account the risks to the environmental and human health. Furthermore, two regulations, 1829 and 1830/2003/EC [101,102], govern the authorization and traceability of food and feed consisting of GMOs or their derivatives. They are subject to extensive, case-by-case, science-based food evaluation by the European Food Safety Authority (EFSA). However, more than half the 28 countries in the European Union (EU) have decided to ban GMO production for food and feed applications, particularly because their risk/benefit balance is perceived as very unfavorable and because the general public often lacks confidence in their promoters and the regulatory process. Despite the ma-

jority of the recent innovations around gene editing in microalgae are coming from groups based in the EU [30], the majority of the commercial applications of these technologies occur outside of the EU, with companies not investing in research and development around this area in the EU, causing an economic stalemate across all sectors.

## 5. Concluding Remarks

The main issues that are currently limiting large-scale production of PUFA-derived microalgae are: (1) the economic unsustainability of some industrial processes, especially those based on the employment of enclosed systems, which are the most suitable systems for the production of genetically modified strains; (2) the inadequacy of open systems for species vulnerable to contamination, temperature, and/or irradiance fluctuations; and (3) a certain reticence to exploit gene-edited microalgae as food or feed supplements. In our opinion, a biorefinery approach aimed at exploiting the whole biomass enhancing the co-production of other commodities and/or minimizing waste is the basic requirement to promote the industrial production of microalgal PUFAs and other high-value compounds. Large-scale production of microalgae to obtain specialty products could be further enhanced by higher confidence in the potential of genetically engineered strains and their official recognition as nutraceuticals, as well as technological advances to completely exclude risks of contamination of aquatic environments with GMOs.

**Author Contributions:** A.S. (Anna Santin), S.B. and A.S. (Angela Sardo): conceptualization and original draft preparation; M.T.R., F.P.E., M.I.F., M.B. and E.C.: review and editing; A.S. (Angela Sardo): supervision. All authors have read and agreed to the published version of the manuscript.

**Funding:** This research received no external funding.

**Institutional Review Board Statement:** Not applicable.

**Informed Consent Statement:** The authors did not require any signed informed consent.

**Data Availability Statement:** Not applicable.

**Acknowledgments:** Anna Santin, Martina Blasio, and Elena Cavalletti were supported by a PhD fellowship funded by the Stazione Zoologica Anton Dohrn (Open University–Stazione Zoologica Anton Dohrn PhD Program).

**Conflicts of Interest:** The authors declare no conflict of interest.

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
