# Peer review of "Microalgae-Based PUFAs for Food and Feed: Current Applications, Future Possibilities, and Constraints"

_jmse, doi:10.3390/jmse10070844_

Round 1
Reviewer 1 Report
In this short review, authors mainly focused on the production of PUFAs from microalgae. They did substantial collection and summary of the microalgal-derived product PUFAS, and also summarized the genetic engineering strategies developed in patents for improving the production of PUFAs in many different types of microalgae. Although many reviews focusing on the aspect of PUFAs have been published, the special attention of their market value is eye-catching, and thus this publication would benefit the community.
One minor revision, Line 80, “Crypthecodinium cohn” or Crypthecodinium cohnii?
Author Response
We thank the reviewer for the positive opinion expressed on this review article, and for pointing out the misprint error. We corrected the name of the dinoflagellate species cited on line 80 (Crypthecodinium cohnii).
Reviewer 2 Report
This article could be interesting for biotechnologists because it give comprehensive information on commercially available microalgae-derived commodities.
Line 80. Misprint Crypthecodinium cohnii,
Line 86. Meteoric?
Author Response
We are grateful for the positive opinion on this article expressed by the referee. We corrected the name of the dinoflagellate species on line 80, and replaced the term meteoric with the more correct one meteorological on line 86.
Reviewer 3 Report
It's an interesting review of research trends for the commercial use of microalgae. Provides important information about existing companies that produce microalgae or microalgae products. In addition, it summarizes the research into genetically modified microalgae for lipid production. The article is well structured and comprehensive.
I consider that a table with comparative data on the productivity of biomass, lipids, and PUFA from different natural species of microalgae and the corresponding genetically modified ones, will significantly upgrade the article.
Author Response
We thank the reviewer for expressing a good opinion on our article, and really appreciate the suggestion to add a table containing more information about the production and productivity of biomass and high-valuable lipids from microalgae. In the new table, we collected information regarding algal strains already cultured to obtain marketable products, since the main focus of this review article is to provide an overview of microalgae-derived lipids which already have a commercial value. We recently published another literature review (https://www.ncbi.nlm.nih.gov/pmc/articles/PMC8707597/pdf/molecules-26-07697.pdf, see Table 2) where we collected information on the effect of genetic modifications of several algal strains from previous scientific papers. However, most of the results described were not exploited for true industrial applications.
In the new table of the present paper, we avoided overlaps with our previous works. Unfortunately, data available in literature are extremely inhomogeneous, and sometimes no information regarding the values of biomass and lipid content are not reported by the authors. However, we did our best to emphasise the potential of some GMOs for industrial applications (the increase of PUFAs, and in general, of the lipid fraction, is, in lots of cases, impressive). We also inserted in the table data of biomass and lipid productivities of some species which are not used as test-organisms for genetic modifications, such as Arthrospira and Tetraselmis spp., but are commonly used for large-scale cultivation and are source of food&feed supplements.